# Comparative Study of the Petal Structure and Fragrance Components of the *Nymphaea hybrid*, a Precious Water Lily

**DOI:** 10.3390/molecules27020408

**Published:** 2022-01-09

**Authors:** Qi Zhou, Man Shi, Huihui Zhang, Zunling Zhu

**Affiliations:** 1College of Environment and Ecology, Jiangsu Open University, Nanjing 210017, China; zhouqi514@njfu.edu.cn; 2Co-Innovation Center for Sustainable Forestry in Southern China, Nanjing Forestry University, Nanjing 210037, China; zhanghuihui@njfu.edu.cn; 3State Key Laboratory of Subtropical Silviculture, Zhejiang Agriculture and Forestry University, Hangzhou 311300, China; shiman1031@126.com; 4College of Landscape Architecture, Nanjing Forestry University, Nanjing 210037, China

**Keywords:** *Nymphaea hybrid*, microstructure, volatile compounds, HS-SPME-GC-MS, multivariate analysis

## Abstract

*Nymphaea hybrid*, a precious water lily, is a widely-cultivated aquatic flower with high ornamental, economic, medicinal, and ecological value; it blooms recurrently and emits a strong fragrance. In the present study, in order to understand the volatile components of *N. hybrid* and its relationship with petals structure characteristics, the morphologies and anatomical structures of the flower petals of *N. hybrid* were investigated, and volatile compounds emitted from the petals were identified. Scanning and transmission electron microscopy were used to describe petal structures, and the volatile constituents were collected using headspace solid-phase microextraction (HS-SPME) fibers and analyzed using gas chromatography coupled with mass spectrometry (GC-MS). The results indicated that the density and degree of protrusion and the number of plastids and osmiophilic matrix granules in the petals play key roles in emitting the fragrance. There were distinct differences in the components and relative contents of volatile compounds among the different strains of *N. hybrid*. In total, 29, 34, 39, and 43 volatile compounds were detected in the cut flower petals of the blue-purple type (Nh-1), pink type (Nh-2), yellow type (Nh-3) and white type (Nh-4) of *N. hybrid* at the flowering stage, with total relative contents of 96.78%, 97.64%, 98.56%, and 96.15%, respectively. Analyses of these volatile components indicated that alkenes, alcohols, and alkanes were the three major types of volatile components in the flower petals of *N. hybrid*. The predominant volatile compounds were benzyl alcohol, pentadecane, trans-α-bergamotene, (E)-β-farnesene, and (6E,9E)-6,9-heptadecadiene, and some of these volatile compounds were terpenes, which varied among the different strains. Moreover, on the basis of hierarchical cluster analysis (HCA) and principal component analysis (PCA), the *N. hybrid* samples were divided into four groups: alcohols were the most important volatile compounds for Nh-4 samples; esters and aldehydes were the predominant volatiles in Nh-3 samples; and ketones and alkenes were important for Nh-2 samples. These compounds contribute to the unique flavors and aromas of the four strains of *N. hybrid*.

## 1. Introduction

Volatile organic compounds (VOCs) are involved in a wide range of functions in ornamental plants [1,2]. For example, they attract pollinators such as insects, which play an important role in the reproduction of plants and enable the development and growth of fruits and seeds [3,4]. VOCs contribute to protection against herbivores, microbial growth, and abiotic stress [5,6]. In some plants, VOCs have allelopathic effects on surrounding plants, protecting the plant and reducing damage from the outside world when they are injured [7,8]. In addition, these compounds improve the aesthetic and economic value of plants, which are important indicators of the ornamental value of cut flowers and plants [9]. Due to the importance of VOCs emitted from plants, research on the floral substances of ornamental plants has gradually attracted attention. This mainly involves the identification of plant volatiles, analyses of the main floral components, and the effects of volatile components on insect pollination behavior [10,11,12]. In addition, with the development of molecular biology, the anabolic pathways of floral substances have been gradually explored, and new methods have been developed for achieving genetic improvements of floral substances [13,14,15].

The main secretory organs of flower fragrances in plants are odorant glands, which are generally distributed in the whole inflorescence of plants, including petals, anthers, calyx, and pedicels, and some volatiles of plants are produced by specific odorant glands [16,17,18]. The odorous glands of plants are usually shaped like whips, brushes, rods, or papillae on the epidermal cells, and they are often shaped like ridges, folds, cones, glandular hairs, or papillae [19]. Scanning electron microscopy (SEM) is often used to observe the structures of plant aroma secretion glands, and transmission electron microscopy (TEM) is used to observe the changes in internal substances in aroma secretion tissues to study the release mechanisms for plant aroma substances [20,21]. Observations of the structures of petal fragments during flowering of *Michelia laba* showed that the granular contents in the petals might be the main precursors to fragrance release in the organs [22]. Studies on the flowers of *Jasminum sambac* have found that there are a large number of granular substances in the upper and lower epidermal cells of its petals. The number of these particles was significantly reduced during opening of the flower, and the particles finally disintegrated. It is speculated that these granular substances were related to the formation of the jasmine fragrance [23].

The aromatic substances in plants are mainly derived from secondary metabolites [24]. At present, more than 1000 floral compounds have been identified in plants, and they cover more than 60 families [25]. Common aromatic substances include linalool, benzyl alcohol, benzaldehyde, basil, phenethyl alcohol, and so on, which are found in many plants [26,27]. It was found that there are great differences in floral compositions, relative contents, and release amounts among different species and even for the same species in different environments [28]. For example, comparing the volatile compositions between the tropical water lily and the cold water lily, researchers detected 79 kinds of volatile compounds in the tropical water lily and 71 compounds in the cold water lily. Among these volatile compounds, 31 were aromatic substances. Compared with the other two types of water lilies, the amounts of volatile substances released by tropical water lilies were higher than those of cold-resistant water lilies [29].

*Nymphaea* L. is a perennial ornamental aquatic flower belonging to the Nymphaea family, which is distributed among tropical, subtropical, and temperate regions [30]. There are approximately 46 species worldwide, which can be divided into two categories: tropical water lilies and cold-resistant water lilies [31]. Among them, *N. hybrid* is a precious species of tropical water lily with gorgeous colors, beautiful postures, and pleasant flower fragrances [32]. In the 1970s, after the introduction of the American fragrant water lily as a breeding material, Chinese horticultural workers carried out crossbreeding for many years and cultivated a number of red, yellow, pink, purple, and other varieties, which were identified as *N. hybrid* by the Nanjing Institute of Comprehensive Utilization of Wild Plants [33,34]. At present, *N. hybrid* is mainly cultivated in Guangzhou, Fujian, Hainan, Zhejiang, Sichuan, Jiangsu, and some northern areas in China. *N. hybrid* is an excellent plant for water beautification and purification [35] and can also be used as high-quality aquatic cut flowers with pleasant aromas enjoyed by many people. Therefore, *N. hybrid* germplasms represent a rich source of aromatic aquatic plants. Moreover, due to the richness of nutrients and bioactive substances in its flowers, numerous studies have been published regarding its broad prospects for applications in foods [32,35], medicine [36,37], cosmetics [38,39], and so on, and it has attracted considerable research interest. Zhang et al. [40] studied the flowering characteristics and reproductive biology of *N. hybrid* and found that their floral traits were well adapted to xenogamy and bee pollination. However, no reports pertaining to the microstructures and compositions of the volatiles produced by various *N. hybrid* flowers have been published, which severely hindered the development and application of the aroma plants.

The objective of the present study was to examine the anatomy and cytology of flower petal structures and identify the volatile components produced by different strains of *N. hybrid* flowers by using HS-SPME-GC-MS analysis to provide basic information on the essential oils that could be exploited and to expand the applications of *N. hybrid* in the fields of horticulture, medicine, and cosmetics.

## 2. Results

### 2.1. Observation of the Microstructure of the Flower Petals of N. hybrid

#### 2.1.1. Basic Structures and Characteristics of Flower Petals

Morphological and ultrastructural observations of *N. hybrid* petals are presented in Figure 1. Using scanning electron microscopy (SEM), it was found that the cells on the surfaces of flower petals were irregular and massive, with abundant folds on the cell wall (Figure 1A). The petaline epidermis has rounded stomata (Figure 1A), and papillae were present on both epidermises of petals (Figure 1B). In cross section, the petal can be divided into three parts: upper and lower epidermis, basic parenchyma cells, and vascular tissues (Figure 1C). Petals comprised 8–9 layers of parenchyma cells. The upper and lower epidermis is a layer of round or oval cells, which are arranged closely and have no obvious differences in morphology. The basic parenchyma cells were loosely arranged, lying or upright, and their shapes and sizes were inconsistent. The thicknesses of the petals ranged from 120 to 150 μm. Using ultrastructural observation of the parenchyma cells, the structures of the cell wall, nucleus, central vacuoles, and cell inclusions were clearly observed by transmission electron microscopy (TEM) (Figure 1D–F). Among them, the plastids were arranged along the cell wall in round or oval shapes, and a small number of starch grains and black oil droplets of different shapes and sizes were seen inside the plastids; these were the secreted osmiophilic matrix granules. It appears that the parenchyma cells of the petals are involved in the production and secretion of the volatile, aromatic compounds emitted from the flowers of *N. hybrid*.

#### 2.1.2. Differences in Cell Structures of *N. hybrid*

There were differences in petal cell structures among the four strains of *N. hybrid* (Figure 2). The shapes of petal epidermal cells of Nh-1 were more irregular than those of the other three strains. The epidermal cells of Nh-3 were broadly ovoid, while those of Nh-2 and Nh-4 were elongated (Figure 2A). There were also differences in the densities of the abundant papillae and degrees of protrusion on the cell walls of petal surfaces for the four different strains of *N. hybrid*. The density and degree of protrusion of Nh-3 were higher than those of the other three strains (Figure 2B).

TEM showed that the ultrastructures and cell inclusions of the petals of the four strains of *N. hybrid* were significantly different (Figure 3). The internal structures of petal epidermal cells and osmiophilic matrix granules were observed. In this experiment, the osmiophilic matrix granules in petal cells of four strains were compared. Nh-3, followed by Nh-4, had the largest numbers of plastids and osmiophilic matrix granules; many secretion structures full of osmiophilic matrix granules were found in their cells, and there were similar pearl strings of osmiophilic matrix granules distributed around their cell walls. Fewer plastids were found in Nh-1 and Nh-2, and there were almost no osmiophilic matrix granules around the cell walls.

### 2.2. Volatile Compounds of the Flower Petals of N. hybrid

#### 2.2.1. Identification and Concentration of the Volatile Compounds in *N. hybrid*

A typical GC-MS total ion chromatogram (TIC) for the volatile chemical profile of *N. hybrid* is shown in Figure 4. Using GC-MS analysis, 64 volatile compounds from the flower petals of four strains of *N. hybrid* were detected and identified (Table 1). The results showed that 29, 34, 39, and 43 volatile compounds were detected in Nh-1, Nh-2, Nh-3, and Nh-4 with total relative contents of 96.78%, 97.64%, 98.56%, and 96.15%, respectively. Some of these compounds were found in only a few of the samples in this study. The volatile components were classified into seven classes, including seven alcohols, six esters, seven aldehydes, six ketones, 13 alkanes, 22 alkenes, and three others. Among these volatile components, terpinen-4-ol, (-)-verbenone, α-ionone, (E)-α-atlantone, and D-limonene were monoterpenes, they only accounted for a small amount. β-Eudesmol, 7-epi-sesquithujene, cedr-8-ene, α-santalene and the other nine alkenes were sesquiterpenes, and they were the major terpenes of volatile components in *N. hybrid*. There were 18 identical compounds found in all four strains of *N. hybrid*. However, the contents of the volatile compound varied markedly among the flowers. Fifteen compounds were detected only in Nh-4, and benzyl alcohol (44.45%) was the most abundant component in this sample; its proportion was significantly higher than those in other samples (*p* < 0.05). In Nh-3, the proportions of total esters (13.28%) and aldehydes (7.22%) were obviously higher than those in the other samples. The levels of 2-heptadecanone (3.00%), α-sesquiphellandrene (3.77%), α-farnesene (2.28%), (-)-zingiberene (2.15%), (6E,9E)-6,9-heptadecadiene (15.76%), and 8-heptadecene (5.66%) were higher in Nh-2 than in other samples, and the differences were significant (*p* < 0.05).

#### 2.2.2. Comparison of the Main Volatile Compounds

We investigated the relative amounts of the main 13 chemical compounds released that were greater than 1% in Nh-1, Nh-2, Nh-3, and Nh-4, with total relative contents of 90.61%, 87.62%, 91.70%, and 85.48%, respectively (Table 2). The levels of the same chemical compound were different in different samples. For example, in Nh-1, benzyl alcohol (32.57%) and pentadecane (22.43%) constituted more than half of the total volatile compounds, and benzyl alcohol has a sweet, flower flavor. No benzyl acetate was detected in Nh-4, but it contained higher levels of benzyl alcohol and trans-α-bergamotene, with a relative content of 55.63%, and trans-α-bergamotene was also a type of sesquiterpene, which had a wood, warm, tea flavor. The relative amount of alkenes released from Nh-2 was much higher than the amounts released by other samples, especially for (6E,9E)-6,9-heptadecadiene and 8-heptadecene; the levels of these compounds were more than four times the levels in Nh-4. The benzyl acetate (12.18%) and benzaldehyde (6.62%) levels in Nh-3 were significantly higher than those in other samples.

#### 2.2.3. Relative Abundances of Different Classes of Volatile Compounds

As shown in Figure 5, the volatile components in flower petals differed significantly among the *N. hybrid* samples. Alkenes, alcohols, and alkanes were the three major types of volatile components contained in the flower petals of *N. hybrid*. Among them, the proportions of alkenes were the largest present in Nh-1, Nh-2, and Nh-3, accounting for 34.04%, 44.63%, and 28.02%, respectively. Nh-4 contained the highest proportion of alcohols, which constituted 46.56% of the total volatile components. However, in Nh-3, the proportions of esters (13.48%) and aldehydes (7.33%) were significantly higher than those in the other three samples.

#### 2.2.4. Relationships among Different Classes of Volatile Compounds

Correlation analysis was used to explore the relationships among different classes of volatile compounds (Table 3). Ketones and alkanes were strongly negatively correlated with total alcohol content. Aldehydes were highly correlated with ester content (r = 0.945, *p* < 0.01) but negatively correlated with the contents of ketones and alkenes. Ketones were significantly correlated with alkanes and alkenes. However, no significant relationships were found among alcohols, esters, and aldehydes.

These correlations can aid the selection of varieties with improved aroma qualities, since selection for one trait may lead to selection for genetically improved traits. The relationships among different volatile compounds indicate the complexity of flower aroma metabolites.

### 2.3. Multivariate Analysis

#### 2.3.1. Hierarchical Cluster Analysis of Volatile Compounds

Hierarchical cluster analyses (HCA) were used to analyze the data for 64 volatile compounds obtained from four strains of *N. hybrid*, and three samples were analyzed for each strain (Figure 6). The 12 samples of *N. hybrid* could be divided into four groups. The first group was Nh-4, including Nh-4a, Nh-4b, and Nh-4c, which were characterized as having high concentrations of benzyl alcohol, dihydrocurcumene, (E)-β-farnesene, trans-α-bergamotene, 2,6,10-trimethyltridecane, cis-3-hexenyl hexanoate, 3-methyltridecane, heptane, and α,3-dimethylstyrene. The second group included the three samples of Nh-2 with high concentrations of tridecane, α-sesquiphellandrene, α-farnesene, (-)-zingiberene, 2-heptadecanone, heptadecane, (6E,9E)-6,9-heptadecadiene, and 8-heptadecene. This strain was characterized by an extremely high alkene content. The third group contained the Nh-1 strain with high concentrations of α-bisabolene, (E)-α-atlantone, heneicosane, α-curcumene, and (E)-γ-bisabolene. The other three samples of Nh-3 were classified into group four. The dominant volatiles in this group were benzyl acetate, benzaldehyde, p-anisaldehyde, guaiazulene, and anisyl acetate.

#### 2.3.2. Principal Component Analysis of Volatile Compounds

The main component analysis was used to explore the data matrix order. This helps to determine the importance of parameters in the total variability through vector size, loads, and respective percent contributions. To identify which volatiles contributed the most to the differences among the strains of *N. hybrid* and for a better description of these scented plants, the data for 64 volatile compounds identified in *N. hybrid* and the seven different classes of volatile compounds were analyzed by using principal component analysis (PCA). In Figure 7A, the first two components of PCA explained 71.8% (PC1) and 21.4% (PC2) of the variation, or 93.2% of the combined variance; in Figure 7B, the first two components of PCA explained 62.9% (PC1) and 31.9% (PC2) of the variation, or 94.8% of the combined variance. Therefore, each of the two principal components was chosen to construct a loading diagram.

Using the factor scores, the tested *N. hybrid* samples were positioned in a two-dimensional space that was divided into four groups based on the scores, and the resulting scatter plot is shown in Figure 7A,B. In Figure 7A, the volatiles that had high positive scores on PC1 included benzyl alcohol, 2-hexyl-1-octanol, butyric acid (E)-3-hexenyl ester, methyl salicylate, cis-3-hexenyl hexanoate, hexyl hexanoate, nonanal, decanal, (Z)-7-hexadecenal α-ionone, and so on, which were highly positively correlated with the Nh-4 samples. Volatiles with high positive scores on PC2 comprised benzyl acetate, anisyl acetate, benzaldehyde, p-anisaldehyde, (6E,9E)-6,9-heptadecadiene, and 4-methylanisole, which were positively correlated with the Nh-3 samples. The group consisting of the three samples of Nh-1, with negative PC2 scores, was only characterized by high amounts of (E)-α-atlantone. The last group consisted of Nh-2 samples with negative PC1 scores, and PC2 scores were characterized by high amounts of terpinen-4-ol, falcarinol, hexadecanal, octadecanal, 2-heptadecanone, α-farnesene, (E)-β-farnesene, α-curcumene α-bisabolene, and others. From Figure 7B we can see that alcohols were the most important volatile compounds for the Nh-4 samples; esters and aldehydes were the predominant volatiles in Nh-3 samples; and ketones and alkenes were important for Nh-2 samples. In general, the PCA results were in accordance with the HCA results.

## 3. Discussion

Volatiles vary from plant to plant, and these changes are closely related to the structures of the flower organs [41]. Some researchers have found aromatic glands or osmophores inside the petals and other floral organs of the plants, which release special fragrances through diffusion [42]. This kind of aromatic gland usually has secretive tissue with thick layers of cells or presents as a droplet shape inside the cells and stores substances for short time periods [43,44]. In this study, it was found that the petal surfaces of *N. hybrid* were pleated and papillary, and the epidermal cells contained starches and osmophilic matrix particles in the flowering period. The petals of *N. hybrid* have aromatic glands, and the parenchyma cells under the epidermis of the petals can form aromatic essential oils and temporarily gather aromatic oil substances, which can overflow through the fold structure of the epidermis cells and mastoid protrusions and release the fragrances of *N. hybrid*. Xu et al. [45] studied the corolla structure and volatile substances of *Jasmin japonicum* and found that the glandular hairs and 8–9 layers of parenchyma cells in the petals formed the structural basis for the specific production of floral fragrance. These folds, papillary, and other diverse secretory epidermal cells on the petals of *N. hybrid* give the scent glands a large contact area with air, which is conducive to the release of volatile substances. 

By comparing the morphologies and ultrastructures of the petals in different strains of *N. hybrid*, it was found that the numbers and morphologies of papillae on the epidermis and the plastids, and osmophilic matrix particles in epidermal cells were different. It was speculated that the osmophilic matrix particles in the petals of *N. hybrid* had certain relationships with the formation of fragrances and provided the material basis for the release of fragrance. According to the previous studies, plastids with plastoglobuli are the main organelles responsible for the production of terpenes, esters, aldehydes, and alcohols [46,47]. In this study, among the different flowers, the degree and density of the mastoid protrusions, the plastids and osmophilic matrix particles in the petals of the yellow *N. hybrid* were higher than those of other strains; such anatomic structures may be the main reason for the higher content of esters and aldehydes in the yellow flower, which makes its aromatic substances more abundant and causes the release of higher quantities. Shi et al. [48] confirmed that osmiophilic matrix particles played a key role in floral formation in a study of *Osmanthus fragrans*.

HS-SPME-GC-MS provides a simple, rapid, and reliable technique for analyzing the volatile compounds in plants [49,50,51]. The volatile compounds of many water lily cultivars have been widely studied, and the volatile components differ [29,52]. Shi et al. [53] analyzed the components of essential oils from fresh flowers of the tropical water lily, and 25, 23, and 25 volatile substances were detected in the volatile oils of tropical *N*. ‘Ruby’, *N*. ‘Daubeniana’ and *N. caerulea*, respectively. The main volatile substances were alkynes, alkanes, alkenes, alcohols, and ketones, among which 8-hexadecyne exhibited the highest content. In our study, 64 volatile compounds were detected and identified in the flower petals of four strains of *N. hybrid*, including seven alcohols, six esters, seven aldehydes, six ketones, 13 alkanes, 22 alkenes and three others. In total, 29, 34, 39, and 43 volatile compounds were detected in Nh-1, Nh-2, Nh-3, and Nh-4. Alkenes, alcohols, and alkanes were the three major types of volatile components, but few alkynes were detected, which contrasted with the research results described above. In addition, some researchers discovered that in the cold-resistant water lily, the main aroma components were alcohols, aldehydes, and ketones, and the most abundant substances were pentadecane, 8-heptadecene, 6,9-heptadecadiene, tridecane, and tetradecane. In the tropical water lily, the main aromatic substances were esters, alcohols, terpenes, aldehydes, and ketones, with high contents of 6,9-heptadecadiene, pentadecane, 8-heptadecadiene, farnesene, and Z,Z-10,12-hexadecadienal. There are obvious differences in the aromas of the two ecotypes of water lilies [29]. The flower aroma of the cold-resistant water lily is influenced by nerolidol and lilac alcohols, with orange and clove flower aromas, while the aroma of tropical water lilies is influenced by ethyl benzoate, acetic acid phenylmethyl ester, and 2-heptadecanone, which provide a fruity and sweet flower aroma [54].

*N. hybrid* is a tropical water lily. By summarizing and comparing the main volatile components in *N. hybrid* with the previously studied *Nymphaea* spp. (Table 4) [52,53,55,56], we found the contents of benzyl alcohol, pentadecane, (E)-β-farnesene, (6E,9E)-6,9-heptadecadiene, and 8-heptadecadiene in different strains of *N. hybrid* were high, which is similar to the main volatile substances found in other tropical water lilies. However, the differences were that in *N. hybrid*, it also contains relatively high levels of benzyl alcohol, benzaldehyde, trans-α-bergamotene, α-sesquiphellandrene, α-curcumene, and α-bisabolene. Previous studies on volatile compounds are often focused on terpenes [2,49,57] and in our study a few monoterpenes (terpinen-4-ol, (-)-verbenone, α-ionone, (E)-α-atlantone, and D-limonene) and several sesquiterpenes, mainly trans-α-bergamotene, α-sesquiphellandrene, (E)-β-farnesene, and α-bisabolene, were found in *N. hybrid*. Terpenes are well known for their contribution to pleasant sensory notes, such as herbaceous, fruity, citrus, and floral scents [58]. All these volatiles compounds may give *N. hybrid* a sweeter fruity floral scent than other water lilies. The unique scent of *N. hybrid* may attract pollinators such as *Bombus* sp., *Xylocopa* s.str. *valga* Gerstaecker, and *Apis cerana* [40]. Although the volatile compounds of many plants of the Nymphaea family have been studied, the information on the microstructure of the flowers has not been investigated before. Both the occurrence of the osmophilic matrix particles and the fragrance of the flowers of *N. hybrid* may have largely been overlooked until now.

By detecting the aromas of the essential oils of red and violet *N. hybrid*, Xu et al. [33] found that its essential oil is relatively elegant and distinctive and suitable for use as a natural flavor additive in foods. In our research, it is worth noting that, compared with other strains of *N. hybrid*, the yellow *N. hybrid* contains more esters and aldehydes, mainly benzyl acetate (12.18%) and benzaldehyde (6.62%). Benzyl acetate has a pleasant fruity flavor, and benzaldehyde has a fruity cherry aroma [59]. Due to the microstructures of the flower petals, total relative contents of volatile substances, and distributions of all classes of volatile substances, the yellow *N. hybrid* had particularly high concentrations of aromatic compounds, which can be distilled to obtain essential oils for applications such as aromatherapy, perfumes, flavorings, and food additives.

Based on this study, alkenes, alcohols, and alkanes are the main volatile compounds responsible for the flower aroma of *N. hybrid*. The contributions of different volatile aroma compounds to flower aromas are affected by the odor activity values, flavor dilution factors, and aroma profiles. Given that the present study only analyzed the relative contents of volatile compounds, further studies are required to determine whether certain compounds act as characteristic aroma compounds in *N. hybrid*, and whether the metabolic differences result from environmental conditions, or gene expression.

## 4. Materials and Methods

### 4.1. Plant Materials

Fresh flowers were obtained at the full flowering period from four widely known strains of *N. hybrid* with different colors: Nh-1 (blue-purple type), Nh-2 (pink type), Nh-3 (yellow type), and Nh-4 (white type) (Figure 8). The *N. hybrid* flowers used for these experiments originated from the *N. hybrid* seedlings base in Yancuo Village (24°47′ N, 116°41′ E), Zhangzhou, Fujiang Province, China. This province has a warm and humid subtropical monsoon climate with sufficient sunshine hours and abundant rainfall. The annual average temperature is 22.0 °C, and the precipitation levels and frost-free periods are 1557 mm and 330 d, respectively. The samples were planted with the same specifications used in the planting pond (length × width × depth: 10 m × 20 m × 0.5 m) with the same cultivation conditions (including soil, fertilization, irrigation, and pest control measures), and they were grown in the planting pond for three years. Three fully expanded flower petals (on the first day after flowering) representing three samples from each strain of *N. hybrid* were collected in the morning (06:00–08:00) and immediately processed.

### 4.2. Microstructural Observations of Petals

#### 4.2.1. Analyses with Scanning Electron Microscopy (SEM)

The anatomical features of petals were analyzed using SEM, according to the method of Marinho et al. [43]. Samples of fresh flower petals (*n* = 3 × 4 individuals, total = 12) of *N. hybrid* were collected from a separate one during the full flowering period. The properly cleaned samples were cut into small pieces (approximately 5 × 5 mm) with a sharp blade. The excised petals were fixed in Karnovsky fixative (4% p-formaldehyde, 5% glutaraldehyde in 0.1 M phosphate buffer, pH 7.2) for 48 h, dehydrated in a graded ethanol series, penetrated with isoamyl acetate aldehyde, and dried in a critical point drying apparatus (K850, Emitech, London, UK). Dried sections were mounted on stubs and coated with gold using an ion sputtering apparatus (E1010, Hitachi, Tokyo, Japan); the surface structures and cross sections of the petals were subsequently observed and photographed under a scanning electron microscope (Quanta 200, FEI, Hillsborough, OR, USA). All the chemicals were bought from Sinopharm Chemical Reagent Co., Ltd., Beijing, China. Three replicates were conducted for each experiment.

#### 4.2.2. Analyses with Transmission Electron Microscopy (TEM)

TEM was used to examine petal ultrastructure in accordance with Xu et al. [45]. The petal samples were cut into fragments of approximately 1 mm^2^. Glutaraldehyde (2.5%) was used for 48 h of fixation in 0.1 M phosphate buffer (pH 7.2). The fragments were washed three times with the phosphate buffer. Then, 3 h postfixation was performed in 1% (*w*/*v*) osmium tetroxide in the same buffer, which was followed by dehydration with a graded series of acetone and embedding in Epon 812 epoxy resin. Sections of 1 μm width for light microscopy and 50 nm for TEM were made with an LKB III ultramicrotome (LKB Instruments Inc., AB Bromma, Stockholm, Sweden). Staining was performed for ultrathin sections with uranyl acetate and lead citrate. The cell ultrastructures of the petals were observed and photographed by transmission electron microscopy (Hitachi HU 12 A; Hitachi High-Technologies, Tokyo, Japan). Three replicates were conducted for each experiment.

### 4.3. Volatiles Analysis

In each of three experimental replicates, 2.0 g of petals of each fresh flower was sampled at random. A manual SPME device with 40 mL vials and PDMS-DVB fibers (65 μm) was used for extraction of flower volatiles (Supelco Inc., Bellefonte, PA, USA). The samples were placed in 40 mL glass vials with flowers occupying approximately two thirds of each vial. To maximize the release of volatile compounds, the SPME fiber was exposed in the upper space of the sealed vial for 50 min at 45 °C to adsorb the analytes. After that, the fiber was directly introduced into the injector of the GC-MS for 3 min at 250 °C to desorb the volatiles.

GC-MS analysis was carried out using a Trace 1300 gas chromatograph coupled with an ISQ-LT mass spectrometer (Thermo Fisher Scientific, Sunnyvale, CA, USA). A DB-5 MS capillary column (5% phenylmethylsiloxane, 30 m × 0.25 mm × 0.25 μm, Thermo Fisher Scientific, Sunnyvale, CA, USA) was employed to conduct the analyses. Helium was used as a carrier gas with a flow rate of 1.0 mL/min. The front inlet was kept at 250 °C in splitless mode. The initial column temperature was maintained at 40 °C for 2 min, programmed up to 110 °C at a rate of 4 °C per min and held for 2 min; it was then raised to 150 °C at a rate of 3 °C/min. This temperature was maintained for 2 min and then increased to 200 °C at a rate of 5 °C/min and held constant for 4 min. The flow rate for split injection was 1.0 mL/min. The mass selective detector was used in EI mode with an ionization voltage of 70 eV, and the scanning range was m/z 33–450. The ion source, transfer line, and quadrupole temperatures were 250 °C.

Retention indices (RIs) were calculated by using the retention times of C7–C30 n-alkanes (BNCC, Beijing, China) that were injected under the same chromatographic conditions [60]. The volatile constituents were identified by comparison of their RI and their mass spectra with the NIST, PubChem, and Pherobase databases [2,61]. The relative content of each component in the total volatiles was calculated according to the ion current peak area normalization method [62]. The spectrum of each compound was analyzed by Xcalibur and NIST 2014 (NIST Database, ChemSW Inc., Fairfield, CA, USA), and compounds were identified according to their retention times and the NIST Database. The spectra of the main peaks were compared with those in the relevant literature [29,33,52,53,63] to determine the chemical composition of the floral substances of *N. hybrid*.

### 4.4. Statistical Analysis

Data were analyzed using SPSS statistical package version 22.0 (IBM Corp., Armonk, NY, USA). Duncan’s multiple range test was used to determine the significant differences between samples (*p* < 0.05). Principal component analysis was carried out using Canoco 4.5 software (Microcomputer Power, Ithaca, NY, USA). Hierarchical clustering and heatmap analyses were performed with TB tools software (https://github.com/CJ-Chen/TBtools/releases, accessed on 18 October 2021).

## 5. Conclusions

To the best of our knowledge, this study first reported the volatile components and the related floral structural characteristics of four widely known different colored strains of *N. hybrid*. We found that the scent emissions from flowers varied among the different strains of *N. hybrid*. The aroma components are dominated by high levels of alkenes, alcohols, and alkanes. According to the microstructure of *N. hybrid* flower petals, it was speculated that the osmophilic matrix particles in the petals of *N. hybrid* had a certain relationship with the formation of the fragrance and were the material basis for the release of the fragrance. These results will have great significance for the development and utilization of *N. hybrid* in the perfume, essential oil, and food industries. It can also be used as an aromatic plant in landscape designs.

## Figures and Tables

**Figure 1 molecules-27-00408-f001:**
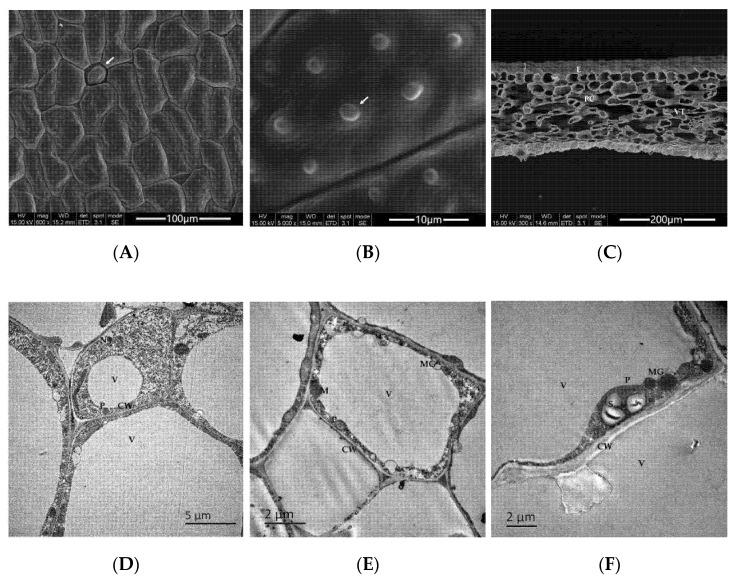
Basic structure of the flower petal of *N. hybrid*. (**A**–**C**) Scanning electron microscopy (SEM) images; (**A**) epidermis cells of the petal, bar = 100 μm; (**B**) papillae on the epidermal cell, bar = 10 μm; (**C**) longitudinal profile of the petal, bar = 200 μm. (**D**–**F**) Transmission electron microscopy (TEM) images; (**D**) cell wall, nucleus, vacuole, and so on, bar = 5 μm; (**E**) mitochondria, osmiophilic matrix granule, plastid, bar = 2 μm; (**F**) starch grains and osmiophilic matrix granules in the cytoplasm, bar = 2 μm. E: epidermis; PC: parenchymal cell; VT: vascular tissue; CW: cell wall; N: cell nucleus; V: vacuole; P: plastid; M: mitochondria; S: starch grain; MG: osmiophilic matrix granule. These same abbreviations are used below.

**Figure 2 molecules-27-00408-f002:**
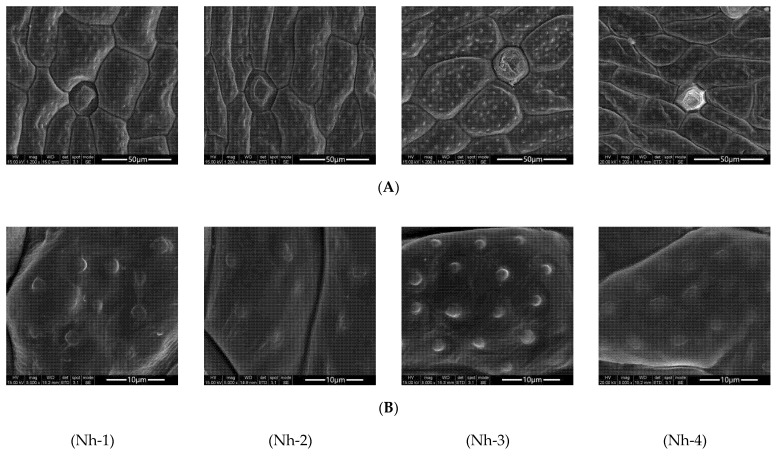
Microcosmic morphologies of the flower petals of *N. hybrid* observed by scanning electron microscopy (SEM). (**A**) Epidermis of flower petals, bar = 50 μm; (**B**) papillae on the epidermis of flower petals, bar = 10 μm. Nh-1, Nh-2, Nh-3, and Nh-4 represent blue-purple, pink, yellow, and white types of *N. hybrid*, as they do in Figure 3 below.

**Figure 3 molecules-27-00408-f003:**
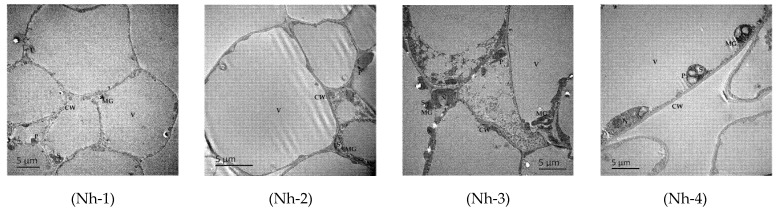
Microcosmic morphologies of the flower petals of *N. hybrid* observed by transmission electron microscopy (TEM), bar = 5 μm.

**Figure 4 molecules-27-00408-f004:**
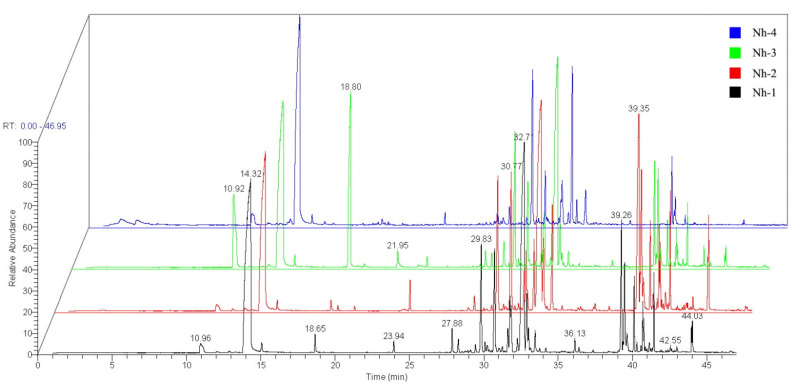
GC-MS total ion chromatograms (TICs) for the volatile chemical profiles of *N. hybrid*.

**Figure 5 molecules-27-00408-f005:**
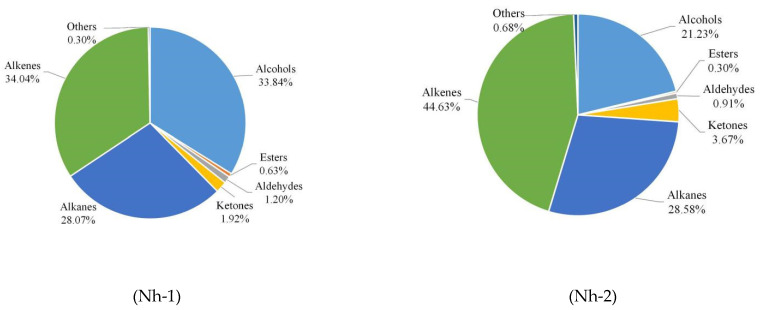
Percentages of alcohols, esters, aldehydes, ketones, alkanes, alkenes, and other volatile compounds released from fully open flower petals of *N. hybrid*.

**Figure 6 molecules-27-00408-f006:**
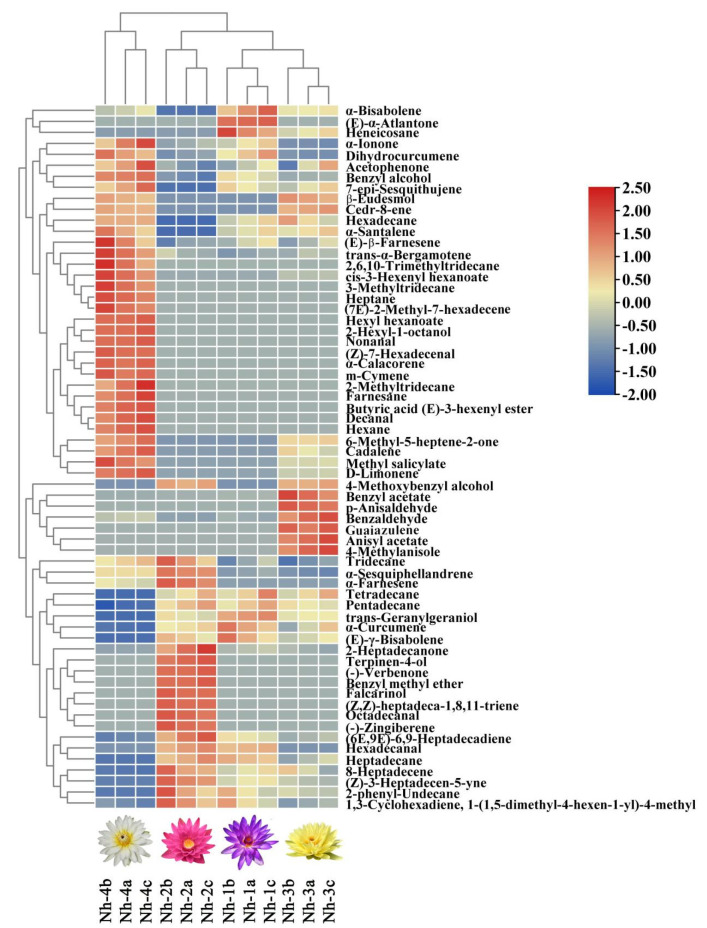
Hierarchical clustering analysis (HCA) and heatmap of volatile compound levels in four strains of *N. hybrid*. Values of all studied volatile compounds per strain are shown on the heatmap on a blue (negative) to red (positive) scale. The HCA and dendrogram of samples were according to Euclidean distances. Nh−1a, Nh−1b, and Nh−1c indicate the three samples of Nh−1, and the same is true of Nh−2, Nh−3, and Nh−4; the same labels are used in Figure 7.

**Figure 7 molecules-27-00408-f007:**
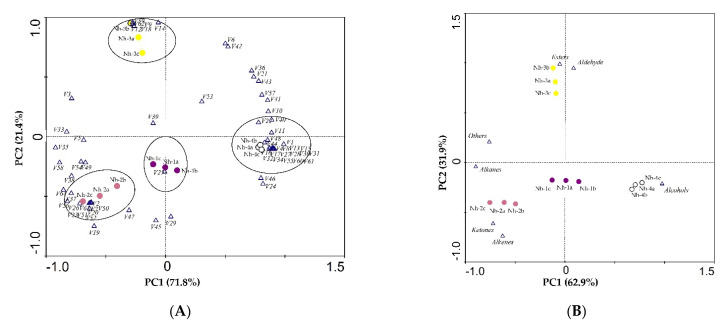
Principal component analysis (PCA) of 12 samples based on PC1 and PC2 scores. V1–V64 represented the volatile substances numbered 1–64 in Table 1. (**A**) Loading plot of samples and the 64 volatile compounds; (**B**) loading plot of samples and the seven different classes of volatile compounds.

**Figure 8 molecules-27-00408-f008:**
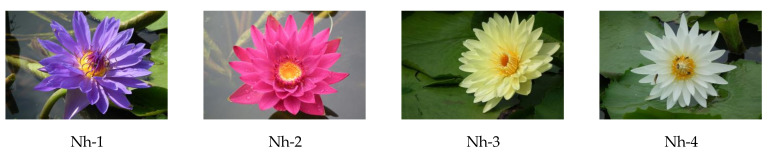
Experimental materials used in this study. Nh-1, Nh-2, Nh-3, and Nh-4 represent the blue-purple, pink, yellow, and white types of *N. hybrid*.

**Table 1 molecules-27-00408-t001:** Volatile compounds released from the flower petals of *N. hybrid*.

Categories	RI (exp.)	RI (lit.)	Volatile Compounds	Molecular Formula	Aroma Description ^1^	Relative Content (%) ^2^			
						Nh-1	Nh-2	Nh-3	Nh-4
Alcohols									
1	1035	1037	Benzyl alcohol	C_7_H_8_O	Sweet, flower	32.57 ^b^	20.25 ^d^	25.27 ^c^	44.55 ^a^
2	1160	1164	Terpinen-4-ol	C_10_H_18_O	Turpentine, nutmeg	nd	0.12 ^a^	nd	nd
3	1245	1247	4-Methoxybenzyl alcohol	C_8_H_10_O_2_	Sweet, powdery creamy	nd	0.12 ^a^	0.11 ^a^	nd
4	1651	1650	β-Eudesmol	C_15_H_26_O	Wood, green	nd	nd	0.12 ^a^	0.11 ^a^
5	2024	2000	Falcarinol	C_17_H_24_O	-	nd	0.12 ^a^	nd	nd
6	2060	2069	2-Hexyl-1-octanol	C_14_H_30_O	-	nd	nd	nd	0.11 ^a^
7	2201	2201	trans-Geranylgeraniol	C_20_H_34_O	-	0.18 ^a^	0.11 ^b^	0.12 ^b^	nd
Total						32.75 ^b^	20.73 ^d^	25.62 ^c^	44.77 ^a^
Esters									
8	1165	1169	Butyric acid (E)-3-hexenyl ester	C_10_H_18_O_2_	-	nd	nd	nd	0.14 ^a^
9	1172	1166	Benzyl acetate	C_9_H_10_O_2_	Fresh, boiled vegetable	0.61 ^b^	0.29 ^b^	12.18 ^a^	nd
10	1195	1193	Methyl salicylate	C_8_H_8_O_3_	Peppermint	nd	nd	0.11 ^b^	0.36 ^a^
11	1380	1370	cis-3-Hexenyl hexanoate	C_12_H_22_O_2_	Fruit, prune	nd	nd	0.11 ^b^	0.77 ^a^
12	1386	1379	Hexyl hexanoate	C_12_H_24_O_2_	Apple peel, peach	nd	nd	nd	0.14 ^a^
13	1415	1413	Anisyl acetate	C_10_H_12_O_3_	Floral, anisic, fruity	nd	nd	0.88 ^a^	nd
Total						0.61 ^b^	0.29 ^b^	13.28 ^a^	1.27 ^b^
Aldehydes									
14	960	963	Benzaldehyde	C_7_H_6_O	Cherry, fruity	0.98 ^b^	0.56 ^c^	6.62 ^a^	1.94 ^b^
15	1081	1083	Nonanal	C_9_H_18_O	Fat, citrus, green	nd	nd	nd	0.11 ^a^
16	1188	1205	Decanal	C_10_H_20_O	Soap, orange peel	nd	nd	nd	0.17 ^a^
17	1255	1252	p-Anisaldehyde	C_8_H_8_O_2_	Mint, sweet	nd	nd	0.60 ^a^	nd
18	1785	1797	Hexadecanal	C_16_H_32_O	Cardboard	0.18 ^b^	0.21 ^a^	nd	nd
19	1798	1798	(Z)-7-Hexadecenal	C_16_H_30_O	-	nd	nd	nd	0.11 ^a^
20	2021	2012	Octadecanal	C_18_H_36_O	-	nd	0.12 ^a^	nd	nd
Total						1.16 ^c^	0.89 ^c^	7.22 ^a^	2.33 ^b^
Ketones									
21	988	975	6-Methyl-5-heptene-2-one	C_8_H_14_O	Citrus	nd	nd	0.12 ^b^	0.19 ^a^
22	1070	1067	Acetophenone	C_8_H_8_O	Must, flower, almond	0.36 ^b^	0.29 ^b^	0.38 ^b^	0.51 ^a^
23	1204	1206	(-)-Verbenone	C_10_H_14_O	Spicy, mint	nd	0.12 ^a^	nd	nd
24	1425	1426	α-Ionone	C_13_H_20_O	Wood, violet	0.27 ^b^	0.17 ^c^	0.12 ^c^	0.41 ^a^
25	1775	1772	(E)-α-Atlantone	C_15_H_22_O	-	0.51 ^a^	nd	nd	nd
26	1882	1883	2-Heptadecanone	C_17_H_34_O	-	0.71 ^b^	3.00 ^a^	0.44 ^b^	0.22 ^b^
Total						1.85 ^b^	3.59 ^a^	1.06 ^c^	1.34 ^b^
Alkanes									
27	600	600	Hexane	C_6_H_14_	Alkane	nd	nd	nd	0.79 ^a^
28	700	700	Heptane	C_7_H_16_	Alkane	nd	nd	nd	0.79 ^a^
29	1300	1300	Tridecane	C_13_H_28_	Alkane	0.40 ^b^	0.86 ^a^	0.31 ^b^	0.71 ^a^
30	1364	1366	2-Methyltridecane	C_14_H_30_	-	nd	nd	nd	0.17 ^a^
31	1371	1373	3-Methyltridecane	C_14_H_30_	-	nd	nd	nd	0.16 ^a^
32	1378	1381	Farnesane	C_15_H_32_	-	nd	nd	nd	0.13 ^a^
33	1405	1400	Tetradecane	C_14_H_30_	Alkane	0.49 ^a^	0.43 ^a^	0.44 ^a^	nd
34	1446	1450	2,6,10-Trimethyltridecane	C_16_H_34_	-	nd	nd	nd	0.23 ^a^
35	1505	1500	Pentadecane	C_15_H_32_	Alkane	22.43 ^a^	23.09 ^a^	20.49 ^a^	11.15 ^b^
36	1600	1600	Hexadecane	C_16_H_34_	Alkane	0.18 ^b^	nd	0.21 ^a^	0.26 ^a^
37	1706	1700	Heptadecane	C_17_H_36_	Alkane	2.64 ^a^	2.85 ^a^	1.41 ^b^	0.39 ^c^
38	1712	1715	2-phenyl-Undecane	C_17_H_28_	-	0.46 ^b^	0.67 ^a^	0.31 ^b^	nd
39	2106	2100	Heneicosane	C_21_H_44_	Alkane	0.57 ^a^	nd	0.25 ^b^	nd
Total						27.17 ^a^	27.91 ^a^	23.43 ^b^	14.80 ^c^
Alkenes									
40	1020	1198	D-Limonene	C_10_H_16_	Citrus, mint	nd	nd	0.11 ^b^	0.45 ^a^
41	1035	1022	m-Cymene	C_10_H_14_	Solvent, gasoline, citrus	nd	nd	nd	0.11 ^a^
42	1392	1393	7-epi-Sesquithujene	C_15_H_24_	-	0.88 ^b^	0.12 ^c^	0.83 ^b^	1.26 ^a^
43	1411	1415	Cedr-8-ene	C_15_H_24_	-	nd	nd	0.12 ^a^	0.11 ^a^
44	1428	1421	α-Santalene	C_15_H_24_	-	0.11 ^b^	nd	0.13 ^b^	0.17 ^a^
45	1440	1434	trans-α-Bergamotene	C_15_H_24_	Wood, warm, tea	4.49 ^b^	5.40 ^b^	5.34 ^b^	10.68 ^a^
46	1448	1448	Dihydrocurcumene	C_15_H_24_	-	0.29 ^a^	0.14 ^b^	0.12 ^b^	0.34 ^a^
47	1455	1456	α-Farnesene	C_15_H_24_	Wood, sweet	nd	2.28 ^a^	nd	0.83 ^b^
48	1476	1480	(*E*)-β-Farnesene	C_15_H_24_	Citrus, green	5.51 ^b^	4.59 ^b^	5.13 ^b^	7.26 ^a^
49	1480	1482	α-Curcumene	C_15_H_22_	-	2.96 ^a^	2.26 ^a^	1.82 ^b^	0.32 ^c^
50	1483	1478	1,3-Cyclohexadiene, 1-(1,5-dimethyl-4-hexen-1-yl)-4-methyl	C_15_H_24_	-	0.76 ^a^	0.98 ^a^	0.42 ^b^	0.30 ^b^
51	1495	1495	(-)-Zingiberene	C_15_H_24_	Spice, fresh, sharp	nd	2.15 ^a^	nd	nd
52	1509	1503	α-Bisabolene	C_15_H_24_	Balsamic	2.69 ^a^	0.13 ^c^	1.73 ^b^	1.39 ^b^
53	1515	1511	(*E*)-γ-Bisabolene	C_15_H_24_	Balsamic	0.22 ^a^	0.18 ^b^	0.13 ^b^	nd
54	1520	1523	α-Sesquiphellandrene	C_15_H_24_	Sweet, fruit, herb	1.18 ^c^	3.77 ^a^	0.80 ^d^	2.62 ^b^
55	1542	1540	α-Calacorene	C_15_H_20_	Wood	nd	nd	nd	0.11 ^a^
56	1665	1655	(*Z*,*Z*)-heptadeca-1,8,11-triene	C_17_H_30_	-	nd	0.15 ^a^	nd	nd
57	1675	1667	(6*E*,9*E*)-6,9-Heptadecadiene	C_17_H_32_	-	9.96 ^b^	15.76 ^a^	6.88 ^c^	3.67 ^d^
58	1680	1671	Cadalene	C_15_H_18_	-	nd	nd	0.23 ^b^	0.52 ^a^
59	1695	1705	8-Heptadecene	C_17_H_34_	-	3.87 ^b^	5.66 ^a^	3.60 ^b^	1.28 ^c^
60	1756	1760	(7*E*)-2-Methyl-7-hexadecene	C_17_H_34_	-	nd	nd	nd	0.21 ^a^
61	1772	1770	Guaiazulene	C_15_H_18_	-	nd	nd	0.23 ^a^	nd
Total						32.94 ^b^	43.57 ^a^	27.62 ^c^	31.64 ^b^
Others									
62	1021	1016	4-Methylanisole	C_8_H_10_O	Naphthyl	nd	nd	0.15 ^a^	nd
63	1391	1396	Benzyl methyl ether	C_8_H_10_O	Fruity	nd	0.11 ^a^	nd	nd
64	1838	1840	(*Z*)-3-Heptadecen-5-yne	C_17_H_30_	-	0.29 ^b^	0.55 ^a^	0.19 ^c^	nd
Total						0.29 ^c^	0.67 ^a^	0.33 ^b^	0.00 ^d^

RI (exp.): Experimental retention indices; RI (lit.): literature retention indices (Pubchem, NIST, and the Pherobase). ^1^ Aroma description was obtained from literature data (http://www.thegoodscentscompany.com/; www.flavornet.org; and pherobase; accessed on 22 December 2021); nd: not detectable. ^2^ Statistical analyses were performed with Duncan’s multiple range test. Means with different letters (a, b, c, d) within a row are significantly different at *p* < 0.05.

**Table 2 molecules-27-00408-t002:** Comparison of the main volatile compounds in *N. hybrid*.

Categories	Volatile Compounds	Relative Content (%) ^1^			
		Nh-1	Nh-2	Nh-3	Nh-4
Alcohols	Benzyl alcohol	32.57 ^b^	20.25 ^d^	25.27 ^c^	44.55 ^a^
Esters	Benzyl acetate	0.61 ^b^	0.29 ^b^	12.18 ^a^	nd
Aldehydes	Benzaldehyde	0.98 ^b^	0.56 ^c^	6.62 ^a^	1.94 ^b^
Ketones	2-Heptadecanone	0.71 ^b^	3.00 ^a^	0.44 ^b^	0.22 ^b^
Alkanes	Pentadecane	22.43 ^a^	23.09 ^a^	20.49 ^a^	11.15 ^b^
	Heptadecane	2.64 ^a^	2.85 ^a^	1.41 ^b^	0.39 ^c^
Alkenes	trans-α-Bergamotene	4.49 ^b^	5.40 ^b^	5.34 ^b^	10.68 ^a^
	α-Sesquiphellandrene	1.18 ^c^	3.77 ^a^	0.80 ^d^	2.62 ^b^
	(E)-β-Farnesene	5.51 ^b^	4.59 ^b^	5.13 ^b^	7.26 ^a^
	α-Curcumene	2.96 ^a^	2.26 ^a^	1.82 ^b^	0.32 ^c^
	α-Bisabolene	2.69 ^a^	0.13 ^c^	1.73 ^b^	1.39 ^b^
	(6E,9E)-6,9-Heptadecadiene	9.96 ^b^	15.76 ^a^	6.88 ^c^	3.67 ^d^
	8-Heptadecene	3.87 ^b^	5.66 ^a^	3.60 ^b^	1.28 ^c^
	Total	90.61 ^a^	87.62 ^b^	91.70 ^a^	85.48 ^b^

^1^ Statistical analyses were performed with Duncan’s multiple range test. Means with different letters (a, b, c, d) within a row are significantly different at *p* < 0.05.

**Table 3 molecules-27-00408-t003:** Linear correlation coefficients among different classes of volatile compounds.

	Alcohols	Esters	Aldehydes	Ketones	Alkanes	Alkenes	Others
Alcohols	1						
Esters	−0.272						
Aldehydes	−0.131	0.945 **	1				
Ketones	−0.579 *	−0.545	−0.629 *	1			
Alkanes	−0.835 **	−0.056	−0.199	0.618 *	1		
Alkenes	−0.461	−0.660 *	−0.699 *	0.934 **	0.465	1	
Others	−0.927 **	−0.043	−0.168	0.005	0.823 **	0.705 **	1

* Significant correlation at *p* < 0.05. ** Significant correlation at *p* < 0.01.

**Table 4 molecules-27-00408-t004:** Comparison of the main volatile compounds in different *Nymphaea* spp.

Categories	Volatile Compounds	Relative Content (%)															
		Nh-1	Nh-2	Nh-3	Nh-4	N1	N2	N3	N4	N5	N6	N7	N8	N9	N10	N11	N12
Alcohols	Benzyl alcohol	32.57	20.25	25.27	44.55	13.0	1.0	4.0	-	17.0	-	-	0.19	7.19	-	-	4.42
	4-Methoxy- benzenemethanol	-	-	-	-	-	-	-	-	-	7.23	-	-	-	-	-	-
	Phytol	-	-	-	-	-	-	-	-	-	13.82	1.34	4.96	5.01	1.40	1.82	-
	Stigmasterol	-	-	-	-	-	-	-	-	-	3.29	-	-	-	-	-	-
	Gamma.-sitosterol	-	-	-	-	-	-	-	-	-	13.19	-	2.49	-	-	-	-
	Stigmasta-5,22-dien-3-ol	-	-	-	-	-	-	-	-	-	-	-	-	2.55	-	-	-
	(Z,E)-Farnesol	-	-	-	-	-	-	-	-	-	-	-	-	-	-	10.81	-
Esters	Benzyl acetate	0.61	0.29	12.18	-	9.0	2.0	3.0	-	4.0	-	-	-	-	-	-	10.42
	2-Methyl-1-butanol acetate	-	-	-	-	-	-	-	-	-	1.84	1.17	1.25	0.42	-	-	-
	n-Dodecyl acetate	-	-	-	-	-	-	-	-	-	-	-	-	-	-	-	-
	Methyl α-linolenate	-	-	-	-	-	-	-	-	-	-	-	-	-	-	1.01	-
	Anisyl acetate	-	-	-	-	3.0	0.4	2.3	-	0.1	-	-	-	-	-	-	-
Aldehydes	Benzaldehyde	0.98	0.56	6.62	1.94	-	-	-	-	-	-	-	-	-	1.01	-	-
	Hexadecanal	-	-	-	-	-	-	-	-	-	5.57	-	0.90	-	-	-	-
	Pentadecanal	-	-	-	-	-	-	-	-	-	-	-	-	-	2.21	-	-
	Anisic aldehyde	-	-	-	-	2.0	-	0.1	3.0	-	-	-	-	-	-	-	-
Ketones	2-Heptadecanone	0.71	3.00	0.44	0.22	-	-	-	-	-	-	-	4.29	3.88	-	-	2.21
	2-Nonadecanone	-	-	-	-	-	-	-	-	-	-	-	-	-	8.24	-	-
Alkanes	Tridecane	-	-	-	-	-	-	-	-	-	-	-	-	-	-	5.66	-
	Pentadecane	22.43	23.09	20.49	11.15	25.0	40.0	32.0	18.0	41.0	-	-	3.11	2.95	11.09	9.73	15.49
	Heptadecane	2.64	2.85	1.41	0.39	-	-	-	-	-	-	1.89	1.42	1.53	-	-	2.23
	Nonane	-	-	-	-	-	-	-	-	-	2.58	1.66	1.78	0.52	-	-	-
	Tetradecane	-	-	-	-	-	-	-	-	-	-	-	-	-	-	10.64	-
	Cyclotetradecane	-	-	-	-	-	-	-	-	-	-	-	2.80	1.71	-	-	-
	Hexadecane	-	-	-	-	-	-	-	-	-	-	-	-	-	2.10	-	-
	2,6,10-Trimethyl-dodecane	-	-	-	-	-	-	-	-	-	-	-	-	-	4.72	-	-
	2-Methyl- bicyclo[2.2.2]octane	-	-	-	-	-	-	-	-	-	-	-	8.66	9.99	-	-	-
	Octadecane	-	-	-	-	-	-	-	-	-	-	-	-	-	5.17	-	-
	Heneicosane	-	-	-	-	-	-	-	-	-	6.60	12.66	7.62	5.00	-	6.67	-
	Tricosane	-	-	-	-	-	-	-	-	-	12.04	-	16.75	-	-	-	-
	Nonadecane	-	-	-	-	-	-	-	-	-	1.04	23.85	2.50	9.54	-	-	-
	Eicosane	-	-	-	-	-	-	-	-	-	2.25	6.86	-	4.10	-	-	-
	Pentacosane	-	-	-	-	-	-	-	-	-	3.33	1.34	6.35	0.07	-	-	-
	Heptacosane	-	-	-	-	-	-	-	-	-	-	5.34	2.99	2.61	-	-	-
	Pentatriacontane	-	-	-	-	-	-	-	-	-	-	1.50	-	-	-	-	-
Alkenes	trans-α-Bergamotene	4.49	5.40	5.34	10.68	-	-	-	-	-	-	-	-	-	1.73	0.46	-
	α-Sesquiphellandrene	1.18	3.77	0.80	2.62	-	-	-	-	-	-	-	-	-	-	-	-
	(E)-β-Farnesene	5.51	4.59	5.13	7.26	-	-	-	-	-	-	-	-	-	5.22	9.25	1.55
	α-Curcumene	2.96	2.26	1.82	0.32	-	-	-	-	-	-	-	-	-	-	-	-
	α-Bisabolene	2.69	0.13	1.73	1.39	-	-	-	-	-	-	-	-	-	-	-	-
	(6E,9E)-6,9-Heptadecadiene	9.96	15.76	6.88	3.67	16.0	21.0	26.0	35.0	4.5	-	-	-	-	-	-	40.1
	8-Heptadecene	3.87	5.66	3.60	1.28	-	-	-	-	-	-	2.96	5.78	6.38	7.95	4.20	15.27
	9-nonadecene	-	-	-	-	-	-	-	-	-	-	0.49	-	2.40	-	-	-
	1,15-Hexadecadiene	-	-	-	-	-	-	-	-	-	-	-	3.16	-	-	-	-
	Squalene	-	-	-	-	-	-	-	-	-	3.34	3.86	4.16	6.05	-	-	-
Others	8-Hexadecyne	-	-	-	-	-	-	-	-	-	-	-	-	-	33.62	30.20	-

-: Not detectable. N1~N12 represent *N*. ‘Bagdad’, *N*. ‘Alice Tricker’, *N*. ‘Mrs. G. Pring’, *N*. ‘Colorate’, *N*. ‘Dauben’, *N*. ‘King of siam’, *N*. ‘Gloriosa’, *N*. ‘Mangala Ubol’, *N. capensis var. Zanzibariansis, N*. ‘Ruby’, *N*. ‘Daubeniana’, and *N. caerulea*, respectively. The data of N1~N5, N6~N9, N10~N11, and N12 were obtained from Mookherjee et al. (1990); Jirapong et al. (2012); Shi, et al. (2017), and Tsai et al. (2019), respectively. The estimation of the relative content in the current study was similar to the data obtained from Jirapong et al. (2012), Shi, et al. (2017), and Tsai et al. (2019). As the literature of Mookherjee (1990) did not introduce the method of data estimation, its data were only for reference.

## Data Availability

Not available.

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
