# Peer review of "Comparative Study of the Petal Structure and Fragrance Components of the Nymphaea hybrid, a Precious Water Lily"

_molecules, 2022, doi:10.3390/molecules27020408_

Round 1

Reviewer 1 Report

General comments

This is an interesting topic related to the improvement the morphologies and anatomical structures of the flower petals of N. hybrid were investigated, and volatile compounds emitted from the petals were identified These results will have great significance for the development and utilization of N. hybrid in the perfume, essential oil and food industries, medicine, and cosmetics.

Therefore, I express my positive opinion for the acceptance of the article to be published in the Molecules after a minor review. Comments are shown below.

Highlights. The highlights are adequate.

Abstract. The abstract is adequate. 

Introduction. This section provides clear and enough information about the research.

Material and methods.

  • Mentions the number of repetitions for each experiment

Results and discussion.

  • Verify the nomenclature of E: epidermis; Pt: parenchymal cell; Vb: vascular tissue; CW: cell wall; N: cell nucleus; V: vacuole; P: plastid; M: mitochondria; S: starch grain; MG: osmiophilic matrix granule. Figures1, 2, and 3 do not show Vb, only V and V1 appear

  • The results and discussion section are adequate

Conclusions.

  • The conclusions are adequate

Tables and figures.

  • Tables and figures are adequate

References. This section is adequate.

Author Response

Response to Reviewer 1 Comments

Point 1: This is an interesting topic related to the improvement the morphologies and anatomical structures of the flower petals of N. hybrid were investigated, and volatile compounds emitted from the petals were identified. These results will have great significance for the development and utilization of N. hybrid in the perfume, essential oil and food industries, medicine, and cosmetics.

Therefore, I express my positive opinion for the acceptance of the article to be published in the Molecules after a minor review. Comments are shown below.

Highlights. The highlights are adequate.

Abstract. The abstract is adequate.

Introduction. This section provides clear and enough information about the research.

Material and methods.

Mentions the number of repetitions for each experiment.

Response: We have added the the number of repetitions for each experiment, the sentence have been rewritten as “Three replicates were conducted for each experiment.”

Results and discussion.

Verify the nomenclature of E: epidermis; Pt: parenchymal cell; Vb: vascular tissue; CW: cell wall; N: cell nucleus; V: vacuole; P: plastid; M: mitochondria; S: starch grain; MG: osmiophilic matrix granule. Figures1, 2, and 3 do not show Vb, only V and V1 appear

The results and discussion section are adequate

Response: We have verified the nomenclature of all the abbreviations, and changed the “Pt: parenchymal cell; Vb: vascular tissue” to “ PC: parenchymal cell; VT: vascular tissue”, and Figures1(C) has been modified accordingly.

Once again, thank you very much for your comments and suggestions.

Reviewer 2 Report

The review “Comparative study of the petal structure and fragrance components of the Nymphaea hybrid, a precious water lily” presents the results on the analysis of petals structure characteristics, morphologies, and anatomical structures of the flower petals of Nymphaea hybrid, along with detection of volatile compounds from its petals. 

The study was well designed and certainly will be interesting for several experts. However, I suggest a complete review of the “chemical approach” of the manuscript. Terpenes are the typical chemical class present in essential oils. Thus, the studies on essential oils are always focused on terpenes. The authors wrote all the text through classification of the chemicals into “alkenes, alcohols, and alkanes”. The authors should mention monoterpenes or sesquiterpenes, instead of alkenes and alkanes through text.
Table 1 contains chemical compounds which were identified in the volatile fraction of the Nymphaea hybrid. Literature data for comparison and unequivocally characterization were missed. 

Author Response

Response to Reviewer 2 Comments

Point 1: The review “Comparative study of the petal structure and fragrance components of the Nymphaea hybrid, a precious water lily” presents the results on the analysis of petals structure characteristics, morphologies, and anatomical structures of the flower petals of Nymphaea hybrid, along with detection of volatile compounds from its petals. 

The study was well designed and certainly will be interesting for several experts. However, I suggest a complete review of the “chemical approach” of the manuscript. Terpenes are the typical chemical class present in essential oils. Thus, the studies on essential oils are always focused on terpenes. The authors wrote all the text through classification of the chemicals into “alkenes, alcohols, and alkanes”. The authors should mention monoterpenes or sesquiterpenes, instead of alkenes and alkanes through text.

Response 1: It is really true as Reviewer suggested that we should mention monoterpenes or sesquiterpenes, instead of alkenes and alkanes through text. According to the Reviewer’s comments, we have added the analysis of terpenes in the the results part and discussion part in the paper.  

Point 2: Table 1 contains chemical compounds which were identified in the volatile fraction of the Nymphaea hybrid. Literature data for comparison and unequivocally characterization were missed. 

Response 2: We have added the literature data for comparison and unequivocally characterization in Table 1. The sentence have been rewritten as “RI (exp.): Experimental retention indices; RI (lit).: Literature retention indices (Pubchem, NIST, and the Pherobase). Aroma description was obtained from literature data (http://www.thegoodscentscompany.com/; www.flavornet.org; and pherobase)”. All the data and aroma description were added in Table 1.

Once again, thank you very much for your comments and suggestions. We have studied the comments carefully and have made correction which we hope meet with approval. The revised portion had been marked up using the “Track Changes” function in revised paper.

Reviewer 3 Report

The manuscript performs a very detailed comparative study of the petal structure and fragrance components of the water lilly, Nymphaea hybrid. Although the introduction is rather long, the study is original, with scientific soundness and modern instrumental techniques. 

My recommendation is accept in its present form. 

Author Response

Response to Reviewer 3 Comments

Point 1: The manuscript performs a very detailed comparative study of the petal structure and fragrance components of the water lilly, Nymphaea hybrid. Although the introduction is rather long, the study is original, with scientific soundness and modern instrumental techniques.

My recommendation is accept in its present form.

Response 1: Special thanks to you for your good comments. We tried our best to improve the manuscript and made some changes in the manuscript. These changes will not influence the content and framework of the paper. And  the revisions had been marked up using the “Track Changes” function in revised paper. We appreciate for the Reviewer’s warm work earnestly, and hope that the correction will meet with approval. Once again, thank you very much for your comments and suggestions.